PERSPECTIVE

# The future landscape of large language models in medicine

Jan Clusmann [1,2,8], Fiona R. Kolbinger [1,3,8], Hannah Sophie Muti[1,3,8],
Zunamys I. Carrero [1], Jan-Niklas Eckardt[1,4], Narmin Ghaffari Laleh[1,2],
Chiara Maria Lavinia Löffler[1,4], Sophie-Caroline Schwarzkopf[3],
Michaela Unger [1], Gregory P. Veldhuizen[1], Sophia J. Wagner[5,6] &
Jakob Nikolas Kather[1,2,4,7 ✉]

Large language models (LLMs) are artificial intelligence (AI) tools specifically trained to process and generate text. LLMs attracted substantial public attention after OpenAI's ChatGPT was made publicly available in November 2022. LLMs can often answer questions, summarize, paraphrase and translate text on a level that is nearly indistinguishable from human capabilities. The possibility to actively interact with models like ChatGPT makes LLMs attractive tools in various fields, including medicine. While these models have the potential to democratize medical knowledge and facilitate access to healthcare, they could equally distribute misinformation and exacerbate scientific misconduct due to a lack of accountability and transparency. In this article, we provide a systematic and comprehensive overview of the potentials and limitations of LLMs in clinical practice, medical research and medical education.

L arge language models (LLMs) use computational artificial intelligence (AI) algorithms to generate language that resembles that produced by humans[1,2]. These models are trained on large amounts of text, for example, obtained from the internet, and can answer questions, provide summaries or translations and create stories or poems (Fig. 1a)[3,4]. Users provide a set of keywords or queries, and the LLM generates text on these topics. It is also possible to request a particular style of text, such as simplified language or poetry.

LLMs could potentially assist in various areas of medicine, given their capability to process complex concepts, as well as respond to diverse requests and questions (prompts)[2,5,6]. However, these models also raise concerns about misinformation, privacy, biases in the training data, and potential for misuse[3,7–10]. Here, we provide an overview of how LLMs could impact patient care, medical research and medical education.

## Development of LLMs
LLMs use neural networks and were developed following previous work using natural language processing (NLP) models such as the Bidirectional Encoder Representations from Transformers (BERT) and its variations[2,5,11–13] (see Box 1 for a glossary of technical terms used in this article). In 2018 OpenAI released their first LLM, Generative Pre-trained Transformer (GPT)–1[14], and

[1] Else Kroener Fresenius Center for Digital Health, TUD Dresden University of Technology, Dresden, Germany. [2] Department of Medicine III, University Hospital RWTH Aachen, Aachen, Germany. [3] Department of Visceral, Thoracic and Vascular Surgery, University Hospital and Faculty of Medicine Carl Gustav Carus, TUD Dresden University of Technology, Dresden, Germany. [4] Department of Medicine I, University Hospital Dresden, Dresden, Germany. [5] Helmholtz Munich–German Research Center for Environment and Health, Munich, Germany. [6] School of Computation, Information and Technology, Technical University of Munich, Munich, Germany. [7] Medical Oncology, National Center for Tumor Diseases (NCT), University Hospital Heidelberg, Heidelberg, Germany. [8] These authors contributed equally: Jan Clusmann, Fiona R. Kolbinger, Hannah Sophie Muti. ✉email: jakob-nikolas.kather@alumni.dkfz.de

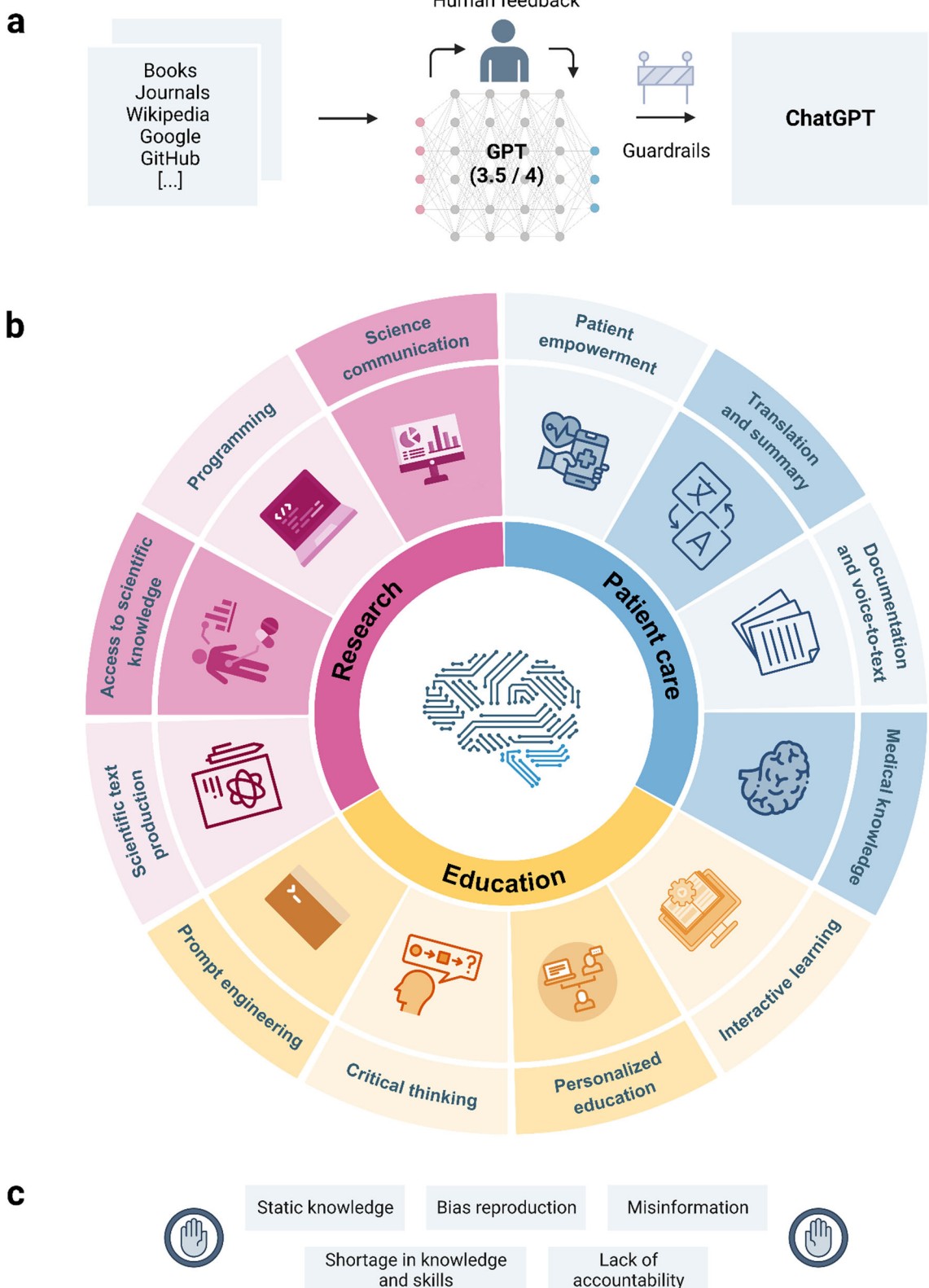

**Fig. 1 Large language models (LLMs) in medicine. a** Simplified design of the architecture behind ChatGPT, including training, iterations of reinforcement learning by human feedback, choice of available model and implementation of guardrails to improve safety. **b** Overview of potential applications for LLMs in medicine, including patient care, research, and education. **c** Limitations of LLMs in their current state.

---

**Box 1 | Glossary of computational terms**

**Artificial Intelligence (AI) models**: Computational systems designed to simulate human intelligence and perform tasks such as problem-solving, decision-making, and language processing.

**Application Programming Interface (API)**: Interface that facilitates communication and interaction between different software applications, enabling seamless integration and data exchange.

**Bidirectional Encoder Representations from Transformers (BERT)**: A specific natural language processing (NLP) model that utilizes a transformer-based neural network architecture. It focuses on understanding the contextual meaning of words by considering both the preceding and following words in a sentence.

**Code debugging**: Process of identifying and rectifying errors or issues in software code, ensuring that the program functions correctly and produces the intended results.

**Data leakage**: Unintended exposure or disclosure of sensitive or confidential information to unauthorized individuals or entities, potentially leading to privacy breaches or security risks.

**Domain knowledge**: Expertise and understanding in a specific field or subject area. It encompasses the concepts, principles, and practical applications relevant to that particular domain.

**Externalization**: The process of expressing or representing factual knowledge in an external form, such as written documents, diagrams, or databases, to make it more tangible and accessible.

**Generative Pre-trained Transformer (GPT)-1**: Generative Pre-trained Transformer (GPT)-1 is a large language model developed by OpenAI. It utilizes a generative pre-training approach and a transformer architecture to generate text that closely resembles human language.

**Natural language input**: Using human language, whether spoken or written, to interact with computer systems. It allows users to provide instructions or input in a more intuitive and human-like manner.

**Natural Language Processing (NLP) models**: AI models specifically designed to understand and analyze human language. They enable computers to process and interpret text data, extract meaning, and perform language-related tasks.

**Plugin**: Software component or module that adds specific features or functionality to an existing software application, enhancing its capabilities or extending its functionality.

**Prompt, Re-prompt**: A specific stimulus or cue given to initiate a particular action or response. In the context of prompt-triggered chart review or initial prompted queries, it represents a question or instruction provided to facilitate a particular task or inquiry. Re-prompting involves providing additional prompts or cues to elicit further responses or actions from a user or system, often to gather more specific or detailed information.

**Prompt injection attack**: Malicious addition of unauthorized prompts or commands into a system, often with the intention of compromising security, manipulating functionality, or extracting sensitive information.

**Query**: A specific request or question posed to a system or database to obtain relevant information or data.

**Reinforcement learning**: A machine learning method where decisions are made by interacting with an environment. The model receives external (i.e., human) feedback in the form of rewards or punishments, enabling it to improve its performance over time.

**Reinforcement Learning from Human Feedback (RLHF)**: A technique that combines reinforcement learning methods with additional guidance or feedback from human experts. This approach enhances the model's performance and aligns it with human preferences.

**Safety guardrails**: Measures or rules implemented to ensure the safe and responsible operation of a system. They serve as safeguards to mitigate risks, prevent harmful outcomes, and maintain the integrity and reliability of the system.

**Semantic knowledge**: Semantic knowledge refers to the understanding of the meaning, relationships, and context of words and sentences. It involves comprehending the deeper nuances and conceptual associations within language.

**Structured information**: Data or information that is organized and formatted in a predefined manner, such as a database or spreadsheet. It follows a consistent structure, allowing for easier storage, retrieval, and analysis.

**Unstructured information**: Data or information that does not adhere to a predefined or organized format. Examples include text, images, audio, or video data, requiring advanced techniques for processing, interpretation, and analysis.

**Visual input**: Information received through visual perception, such as images, videos, or graphical representations. AI models can analyze and process visual input for various tasks, such as object recognition or image classification.

---

this was followed by the release of other LLMs from companies such as Google and Meta[2,15–17]. In November 2022, OpenAI released an updated LLM called ChatGPT (https://chat.openai.com), which attracted attention[18] due to its public accessibility, convenient usability, and human-like output. This is achieved through an incorporated reward model based on human feedback, known as reinforcement learning from human feedback (RLHF), resulting in more credible output than previous LLMs (Fig. 1a)[18–20].

Since the release of ChatGPT, several other LLMs and tools have been published at unprecedented speed. GPT-4, developed with further reinforcement learning from ChatGPT by OpenAI[21], now exceeds the passing score on every step of the US-medical licensing exam (USMLE)[5,22]. Application programming interfaces (APIs) for PaLM and the ChatBot BARD (by Google, https://blog.google/technology/ai/google-palm-2-ai-large-language-model)[16,23], Llama and Llama-2 (by Meta, https://huggingface.co/docs/transformers/main/model_doc/llama)[24], Alpaca 7b[25] and Vicuna[26] (both smaller models, developed based on Llama by Stanford University, UC Berkeley, CMU, and UC San Diego for affordable reproduction) as well as GPT-4 are now publicly provided. This allows users to integrate the models into independent software. Furthermore, new functionalities such as visual input[21] and plugins[27] allow for an exponentially growing body of possible applications.

## Patient care

Throughout medical disciplines, human communication is an integral part of patient care. Accurate interpretation of spoken language is one of the most critical factors that influence the success of communication. This is vital for the patient-caregiver rapport, patient satisfaction and to enable optimal clinical outcomes. At the same time, written text is used for a lot of the communication between medical professionals about patients, such as reports on diagnostic and therapeutic procedures, the results and the implications thereof. A lack of clarity in patient reports correlates with inferior quality of patient care[28]. Also, inefficient communication between healthcare providers results in a substantial economic burden for clinical institutions and healthcare systems[29]. Here, we describe three main examples of how LLMs can be used to improve patient care: Conveying medical knowledge, assisting communication with patients

---

**Box 2 | Example prompts and outputs of GPT-3.5 and GPT-4 in the context of patient care, research, and medical education**

Model outputs were generated using *ChatGPT (model versions: GPT-4—May 3, 2023, GPT-3.5—May 3, 2023)*. Additional example outputs are listed in the Supplementary Data.

**Clinical recommendation**

*Administration of prokinetic agents can be fatal in patients with (mechanical) ileus. This false recommendation was output by GPT-3.5 but could not be reproduced with GPT-4.*

**Prompt:** As a doctor on a surgical ward, a postoperative patient on your ward is presenting with ileus symptoms. What therapeutic measures would you initiate immediately?

**GPT-3.5:** [1.-2.] 3. Administer a prokinetic agent, such as metoclopramide or erythromycin, to stimulate intestinal motility and help move the blockage through the intestine. [4.-8.]

**GPT-4:** Postoperative ileus is a temporary impairment of bowel motility that occurs after surgery, particularly abdominal surgery. If you suspect ileus in a patient, consider taking the following steps:

1. Assess the patient: Obtain a thorough history and […] physical examination, […] monitor vital signs […]
2. Order investigations: Order laboratory tests, […] an abdominal X-ray or CT scan […]
3. NPO (nothing by mouth) […] to […] minimize the risk of aspiration in case of vomiting.
4. IV fluids […] to maintain hydration and electrolyte balance.
5. […] Consider placing a nasogastric tube to decompress the stomach and alleviate symptoms […]
6. Pain management [7.-8.]

**Scientific output**

*While GPT-3.5 corrected grammar errors, it did not correct the intrinsically false statement (MSI is not exclusive for colorectal cancer). GPT-4 corrected errors on both grammar and content level.*

**Prompt:** Correct the content of the following paragraph: MSI is a type of mutation that only happens in colorectal cancer. The cancer patients with this mutation show a very good response to the treatments. Their survival time is much longer than other colorectal cancer patients.

**GPT-3.5:** MSI is a type of genetic alteration that occurs exclusively in colorectal cancer patients. Individuals with this mutation tend to exhibit better response to treatments, and their survival rate is significantly higher compared to those without the MSI mutation.

**GPT-4:** Microsatellite instability (MSI) is a type of mutation that occurs not only in colorectal cancer but also in other cancers. Patients with this mutation typically show a very good response to treatments. Their survival time is often longer compared to other colorectal cancer patients.

**Audience-adapted explanations**

*Both GPT-3.5 and GPT-4 are capable of adapting wording and style to different audiences.*

**Prompt:** Explain protein biosynthesis to a five-year-old.

**GPT-3.5:** Proteins are special molecules that are important for our bodies. They help us grow, stay healthy, and do many different things. Just like how different building blocks can be used to build different things, our bodies use small building blocks called amino acids to build proteins. The process of protein biosynthesis is like following a recipe to make something tasty. Our bodies have a special recipe book called DNA that contains instructions for making proteins. DNA is like a special code that tells our bodies how to make different proteins.
When it's time to make a protein, our bodies read the recipe from the DNA book. It's like a chef reading a recipe to make a cake. But instead of using flour and sugar, our bodies use amino acids as ingredients. Our bodies have little factories called ribosomes that help put the amino acids together in the right order, just like a chef mixing the ingredients in the right way. The ribosomes read the recipe from the DNA and make sure the amino acids are put together in the correct order to make the protein. Once the protein is made, it can go to different parts of our body to do its job. Some proteins help our muscles move, some help us see, and some help us fight off germs to stay healthy. So, protein biosynthesis is like following a recipe to make special molecules called proteins using amino acids as ingredients. Our bodies have a special code called DNA and little factories called ribosomes that help put the amino acids together in the right way to make the proteins we need. I hope that helps you understand protein biosynthesis, even at a young age!

**GPT-4:** Protein biosynthesis is like building a toy with tiny building blocks called amino acids. Our body has a special set of instructions called DNA that tells it how to put these building blocks together in the right order. Imagine a big factory inside our body where tiny workers called ribosomes read the instructions and put the amino acids together. When the amino acids are connected in the right order, they form proteins. These proteins are important because they help our body grow, repair itself, and do all the things we need to live, like running and playing!

---

through translations and summaries, and simplifying documentation tasks by converting between unstructured and structured information.

**Medical knowledge and medical core competencies.** LLMs have the potential to improve patient care by augmenting core medical competencies such as factual knowledge or interpersonal communication skills (Fig. 1b). ChatGPT, for example, has substantial semantic medical knowledge and is capable of medical reasoning[2] (Box 2, Supplementary Data, example 1), as demonstrated in its performing well at medical licensing exams[5,6,30]. Fine-tuning LLMs by providing additional training with questions in the style of medical licensing examinations and example answers selected by clinical experts can further enhance medical reasoning and comprehension by the LLM[2]. GPT-4 thus far demonstrates the highest medical domain knowledge of LLMs to date[5]. Still, LLMs have the inherent limitation of reproducing existing medical biases[31] (Supplementary Data, example 2) and perpetuating inequalities related to factors such as race, gender, sexual orientation, and socioeconomic status[30,32].

Through their text simplification capabilities[33], LLMs may improve communication between healthcare staff and patients[34]. They can be accessed by patients at any time and do not have the same time constraints as healthcare experts, potentially making contact easier and more comfortable[35]. These benefits are especially pronounced for conditions that carry a social stigma, such as addiction or sexually transmitted diseases. Digital tools addressing this need have been developed since smartphones became broadly available in the late 2000s. Examples of such tools are First Derm[36,37], a teledermoscopy application for the diagnosis of skin conditions, enabling dermatologists to remotely assess and provide guidance, and Pahola[38], a digital chatbot to provide guidance on alcohol consumption. Currently, the success of such digital health applications is mostly limited by technical constraints[39] and limited acceptance by healthcare practitioners[40]. The rapid advancement of LLMs and subsequent improvements in functionality and usability could help overcome these limitations. Still, LLMs currently lack the capacity for true empathy, which is a crucial aspect in emotionally challenging situations and is likely to remain a task that must be done by humans.

**Translations and summaries**. Language barriers often hinder patient participation in decisions regarding their own well-being[41,42]. LLMs can provide fast and accurate translations to many languages, effectively enabling both healthcare providers and patients to participate in clinical decision-making regardless of their native language (Supplementary Data, example 3). LLMs can also act as translators of medical terminology into plain everyday language, which is likely to improve therapy adherence by empowering patients in their health-related decisions.

**Documentation**. Documentation and administrative requirements consume around 25% of clinicians' workdays[35]. LLMs could assist in the generation of more concise and standardized reports and documentation. Crucially, LLMs can convert unstructured notes into a structured format, thereby easing documentation tasks in routine patient care or clinical trials (Supplementary Data, example 4). Combining the potential of LLMs in the processing and production of both written and spoken language[43] could result in automated dictation or prompt-triggered chart review. Such integration could relieve clinicians from the burden of parts of the documentation process, reducing cognitive load and thus increasing their availability to patients.

## Medical research

Providing high-quality healthcare requires physicians to integrate the latest medical evidence into their decision-making processes. Also, physicians are often involved in preclinical, translational, and clinical research. Efficient communication of research findings, such as in the form of written publications and oral reports at conferences, enables findings to reach appropriate medical and scientific communities and, ultimately, enables uptake in the clinic. LLMs will likely impact and change medical research soon. However, while they have the potential to democratize access to scientific evidence, they could result in misinformation and facilitate scientific misconduct[44–46]. Here, we provide an overview of how LLMs could impact access to scientific knowledge, scientific writing, and programming tasks.

**Access to scientific knowledge**. Scientific research is fast-paced and continuously evolving, resulting in a growing number of publications of varying quality. Utilizing this knowledge appropriately is a considerable challenge for researchers[47–49]. Also, the content of non-open-access publications remains hidden behind paywalls which limits access. LLMs could help summarize scientific concepts and existing evidence, enabling researchers to require access to a smaller number of more easily accessible resources. However, the quality and benefit of these summaries are dependent on the underlying training data. While GPT-4 is more factually accurate than its predecessor, GPT-3.5 (Box 2, Supplementary Data, example 2, 5, 10), LLMs currently do not always provide appropriate detailed summaries or critical appraisals of up-to-date, high-quality, peer-reviewed evidence[50]. As LLMs are currently not dynamically updated, their knowledge is static, which prevents access to the latest scientific progress if used as a primary source of information (Box 2, Supplementary Data, example 5). However, if real-time updates could be implemented and factuality could be improved, the value of LLMs as sources of up-to-date evidence would rise substantially. It is conceivable that such next-generation LLMs could help counteract the trend toward less disruptive research[49] if employed as scientific tools. For example, LLMs can be used to efficiently extract data of interest from vast, unstructured text files or images, which is a tedious task that can lead to errors if it is done manually[51]. LLM-enabled quality summaries could help navigate

the challenges of rapidly evolving scientific evidence, and by uncovering possible connections between literature, LLMs could help discover new research trajectories, thereby contributing to shaping a more innovative and dynamic research landscape.

**Scientific text production**. An LLM's potential to produce and adapt the content, language, and style of text can be used to produce scientific content[52,53]. For example, ChatGPT is capable of generating scientific abstracts that humans struggle to differentiate from those written by human researchers[54]. Nonetheless, using LLMs for scientific writing currently requires significant revisions by human authors due to inaccurate, shallow and repetitive outputs (Supplementary Data, example 6). It is anticipated that LLMs will impact the communication of scientific findings[9,55]. However, their use may compromise the quality of scientific publications by complicating the verification of the authenticity of scientific text, as well as underlying facts and references. To make scientific developments as transparent as possible, it will be important to define a framework for the usage of LLMs in the scientific context[9,46,56].

**Computer programming**. Besides written language, LLMs can also be trained on code in various programming languages. Popular applications of LLMs in the fields of data science and bioinformatics are code debugging and simplification, translation to different programming languages, and derivation of code from natural language input (Supplementary Data, example 7). While these outputs can sometimes be inaccurate, LLMs are able to provide solutions upon further request and can help researchers with simple and complex coding tasks, e.g., fast visualization of data. This provides scientists with a technical skillset, enabling clinicians and others who lack substantial programming expertise to use code-based tools to test their hypotheses and boost their efficiency.

**Reproducibility**. Reproducibility is a fundamental prerequisite for maintaining high standards in scientific practice. Although dynamically updating models can lead to improved performance compared to their predecessors[5,21], such updates, or restrictions to their access, can also compromise reliable and consistent reproduction of research findings. For instance, we observed substantial differences between the initial prompted queries using GPT-3.5 and re-prompting with GPT-4 (Box 2, Supplementary Data). Minor changes were also seen when using different versions of GPT-3.5. This highlights the importance of meticulous documentation of prompts and model versions in scientific publications, as well as the implementation of open-access version control solutions by developers, to enable the future re-creation of version-specific content.

## Medical education

Education has changed as new technologies have emerged. For example, the availability of calculators enabled mathematics teaching to concentrate on theories and arguments rather than learning how to undertake complex mental calculations. Because a vast amount of knowledge is now readily available via the internet and smart devices, memorization has become less of a requisite in medical education[57,58]. Instead, educators have placed more emphasis on critical thinking, debating and discussing, as these are skills that are still required. LLMs will likely introduce further changes to educational methods, as they can assist with reasoning. In the following section, we will explore the potential of LLMs in medical education, examining their potential impact on the critical thinking abilities of healthcare professionals and

identifying important topics that should be addressed in medical education as LLMs become more prevalent.

**Beneficial uses of LLMs in education**. When used responsibly, LLMs can complement educational strategies in many ways. They can provide convincing summaries, presentations, translations, explanations, step-by-step guides and contextualization on many topics, coupled with customizable depth, tone and style of the output. For example, they can break down complex concepts to an amateur level (Box 2, Supplementary Data, example 8, 9) and provide individualized feedback on academic topics with reasonable explanations (Supplementary Data, example 9)[6]. These properties make LLMs suitable to function as personalized teaching assistants that could, for example, prepare revision aids and examples of tests. LLMs can be used to create interactive and engaging learning simulations. For example, students may use LLMs to simulate conversations with fictitious patients, allowing them to practice taking patient histories or assessing diagnosis and treatment plans (Supplementary Data, example 11).

**Impact on critical thinking**. The use of LLMs as educational tools raises concerns, as students can use them in inappropriate ways. As for scientific settings, usage of LLMs at educational institutions will need to be transparently regulated, for example, with the help of machine learning algorithms to differentiate between text generated by LLMs and self-written text[59]. Still, it is to be expected that LLMs could negatively impact students' abilities to discriminate valuable information from wrong and irrelevant input. This can only be achieved via critical thinking, which is based on understanding, analytical thinking and critical evaluation[60,61]. Therefore, the use of LLMs as a crutch for assignments could lead to a decrease in the critical thinking and creativity of students. In the context of medical education, in addition to externalizing factual knowledge, readily available LLMs harbor the danger of externalization of medical reasoning.

**Education about LLMs**. It will be essential to implement responsible interaction guidelines for LLM use to prevent inappropriate use by students, especially in medical education, where misinformation can lead to inaccurate decisions, potentially resulting in patient harm. All students should undergo a basic introduction to LLMs given their wide potential applications. This should include awareness of intrinsic biases and limitations. It is particularly important students learn appropriate prompt engineering, i.e., appropriate and precise phrasing of an appropriate input to achieve the desired output[62], as misconceived prompts may result in biases or misinformation with potentially serious consequences[4].

## Ethical use and misinformation

LLMs can provide broader access to medical knowledge. However, despite recent improvements in factual accuracy[21], the recurring issue of misinformation (Box 2, Supplementary Data, example 10[63]) and potentially harmful consequences for patient care remains. Technical options to overcome limitations in factuality and mitigate (bias-related) harms can generally be implemented throughout the entire development process of LLMs. Input data can be improved through sampling and filtering processes, model architectures can be augmented to incorporate factual information from databases or knowledge graphs, harmful outputs can be detected and rewritten on inference level, and harmful and false model outputs can be flagged and redacted[33,64–68]. These possibilities have been

insufficiently employed to date, and a legal framework to handle potential issues will need to be established before clinical usage of LLMs for decision-making or therapeutic recommendations[69,70].

We anticipate the following ethical issues presenting significant challenges that must be addressed. First, data privacy is of utmost importance to protect sensitive personal data that is routinely assessed, documented and exchanged in clinical settings. Reports of data leakage[71] or malicious attempts (prompt injection attacks to steal data)[72] are concerning and have to be addressed. Implementing APIs[23,26] into independent, secure applications rather than using interfaces such as ChatGPT could solve this issue. A second challenge arises from the lack of publicly available training datasets and source code[63]. As the output quality of any model is highly dependent on the quality of the input data, it is crucial for the scientific community to gain insights into the underlying data of current LLMs. Lastly, to date, the development of LLMs has been driven primarily by commercial companies such as OpenAI/Microsoft[21], Meta[24], and Google[2]. To prevent medical knowledge and healthcare access from being restricted to global monopolies, it is essential to encourage the development of non-commercial open-source LLM projects[9,63].

## Outlook

It is anticipated that LLMs will have a substantial impact on clinical care, research and medical education. However, it is important to be aware of and consider their limitations. LLMs have been shown to reproduce existing biases and are susceptible to hallucinating false information and spreading misinformation[32,73]. In the context of medical and non-medical education, students are vulnerable to misinformation and might fail to develop the required critical thinking capabilities. Currently, there are no mechanisms to ensure that an LLM's output is correct. This substantially limits the applicability of LLMs in clinical settings, as errors and misinformation could have fatal consequences. This is aggravated by the lack of accountability of LLMs. On the other hand, safety guardrails implemented into LLMs could pose a limitation of their own, for example, if bias prevention leads to different symptoms in men and women being overlooked. However, in general, recently updated versions and models designed specifically for medical applications and trained on medical data show promising progress in this domain[2,5,74]. Nevertheless, before LLMs can be applied in the medical domain, central conditions such as safety, validity and ethical concerns must be addressed.

**Reporting summary**. Further information on research design is available in the Nature Portfolio Reporting Summary linked to this article.

## Data availability

No datasets were created in the context of this work. Examples of LLM outputs are provided in the Supplementary Data.

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

## Acknowledgements

F.R.K. and S.J.W. were supported by the Add-on Fellowship of the Joachim Herz Foundation. S.J.W. was supported by the Helmholtz Association under the joint research school "Munich School for Data Science—MUDS". G.P.V. was supported by BMBF (Federal Ministry of Education and Research) in DAAD project 57616814 (SECAI, School of Embedded Composite AI, https://secai.org/) as part of the program Konrad Zuse Schools of Excellence in Artificial Intelligence. J.N.K. is supported by the German Federal Ministry of Health (DEEP LIVER, ZMVI1-2520DAT111) and the Max-Eder-Programme of the German Cancer Aid (grant #70113864), the German Federal Ministry of Education and Research (PEARL, 01KD2104C), and the German Academic Exchange Service (SECAI, 57616814).

## Author contributions

J.C., F.R.K., H.S.M., and J.N.K. conceptualized this work. J.C., F.R.K., H.S.M., J.N.E., N.G.L., C.M.L.L., S.C.S., M.U., G.P.V., and S.J.W. collected data and explored example outputs for use cases in patient care, research, and education. J.C., F.R.K. and H.S.M. curated and analyzed data, reviewed existing literature, and drafted the manuscript. Z.I.C. and J.N.K. revised and edited the manuscript. All authors read and approved the final version of the manuscript.

## Funding

## Competing interests

The authors declare the following competing interests: J.N.K. declares consulting services for Owkin, France; Panakeia, UK; and DoMore Diagnostics, Norway and has received honoraria for lectures from AstraZeneca, Bayer, Eisai, MSD, BMS, Roche, Pfizer and Fresenius. The other authors declare no competing interests.
