## [Peer Review File · Communications Medicine]

Reviewers' comments:

Reviewer #1 (Remarks to the Author):

This research is a worthwhile effort, but the depth of analysis, language used (and novelty) would need to be improved. The paper needs a deeper analysis for a systematic overview (a Google Scholar search for "ChatGPT" "Medicine" returns 389 results for 2023 alone .. just some of the aspects that are not covered: Ethics, Translational medicine, the possibility of Copyright issues) and the language used would need to be more domain specific for a LLM, not so much "layperson"; for example, terms such as generative AI models, downstream tasks, Reinforcement Learning from Human Feedback should be used.

[Editor's note: given the audience for this will be clinicians we would prefer the layperson language be retained, however, more specific terms should be added for the benefit of any computational experts reading this, perhaps by adding technical terms in brackets after the lay person term?].

Also, at least one very similar article is available online as preprint on medXiv, namely "The Utility of ChatGPT as an Example of Large Language Models in Healthcare Education, Research and Practice: Systematic Review on the Future Perspectives and Potential Limitations" (it should be noted that it was posted on Feb 21 2023, a few days after the authors submitted the paper).

Some additional comments:

- in the section Medical knowledge, it would make sense if the authors focus more on possible misinformation, as it is of high importance;
- Supplementary Table 1: it would make sense to have an additional column where the authors evaluate the output;
- Supplementary Table 1, point nr. 7 (code production and debugging) - the example provided is from a practical perspective useless - ChatGPT can produce concrete code, not just a vague description, which can not be used as it is, for example a simple google search would produce a much more practical solution.

Reviewer #2 (Remarks to the Author):

This is an opinion paper about the future of medicine with the advent of large language models. The authors discuss some challenges and opportunities of LLMs.

In medicine, we have the opportunity to use domain specific LMs, pre-trained language models such as ClinicalBERT, SciBERT, BioBERT to obtain better performance in clinical language applications (summarisation, information extraction, etc.).

A major Challenge of ChatGPT is factuality. This is currently addressed by the AI community since lack of factuality is dangerous in a clinical setting.

There is a lot of research in clinical informatics about these topics.

Supplementary material with use cases are very useful. A more in depth analysis and expansion of this position paper will be immensely useful.

Reviewer #3 (Remarks to the Author):

The manuscript provides an overview on the use of large language models (LLMs) in medicine, which is specially relevant given the recent appearance of ChatGPT. The manuscript focuses on applications of LLMs in medicine across three dimensions: patient care, research, and education.

The manuscript addresses an interesting and timely topic (or, even better, a 'hot topic'). The structure of the work is adequate and the overall idea is very interesting, however I have a few concerns for 2 out of the 3 main topics (patient care, research, education). The discussion for LLMs in research and to some extent also in education seem rather general and not very novel. In other words, if this paper had been about the application of LLMs in the financial sector, these two sections would still apply without much change. In patient care, however, many interesting aspects are discussed, including conversing with the patient when the healthcare worker is unavailable, translating and/or summarising medical data to patients in a way that medical jargon is accessible to a layperson, and helping healthcare workers better do their administrative tasks by using speech recognition and LLMs to extract structured variables from unstructured medical texts. These discussions are specific to medicine and are very important.

I suggest that the authors revise their manuscript to include the main shortcomings and promises of LLMs for research and education *in medicine* rather than in general science (see [1,2,3,4,5], some already included as citations in your manuscript, for examples of LLMs in science in general, instead of specifically in medicine). Perhaps better highlighting what contributions are specific to medicine vs general for science could be a way to contextualise your manuscript's within the literature.

A few specific comments/questions:

- "Large Language Models (LLMs) are artificial intelligence (AI) tools specifically trained to interpret and generate text." -> I would not say 'interpret' since (to me) it implies critical thinking, I would rather refer to it as 'process', since this is the term typically used in the literature to refer to models that make predictions using natural language.

- "LLMs improve communication between healthcare staff and patients." -> I would rephrase it as "LLMs may improve communication between healthcare staff and patients", especially if there are no citations to substantiate this claim.

[1] <https://www.nature.com/articles/d41586-023-00288-7>

[2] <https://www.nature.com/articles/d41586-023-00340-6>

[3] <https://www.nature.com/articles/d41586-023-00107-z>

[4] <https://www.nature.com/articles/d41586-023-00191-1>

[5] <https://www.nature.com/articles/d41586-022-03479-w>

Reviewer #4 (Remarks to the Author):

Summary

The authors present a timely perspective of the current state of Large language models (LLM) and present a vision and use cases for how such technology (given current or near term capabilities) may impact patient care and medical research. Most importantly, the paper is written with the goal of being accessible to a general audience inclusive of researchers and clinicians within medicine and

the biological sciences. Given the amount of sensationalism and misinformation surrounding LLM, I believe this goal is critically important and worthy of publication. This manuscript provides much of this needed context, as well as providing a great source for references and further reading. With this in mind, my comments below are less directed at dissecting the article's publication value, which I support, but more suggestions and perspectives on potential improvements/additional considerations

Major Comments

I think all the use cases are worthwhile thinking about, but I think it should be made very clear that submission of protected health information to the currently available LLMs should not be advocated. While there is an option to request no data submitted be used for future model development, there is no way of verifying this assertion. Given how easy and tempting it may be to use these LLM such as ChatGPT as note writing or diagnostic reasoning tools, the possibility of information leakage into the public domain through these LLM is real and may be worth explicitly mentioning for the sake of clinician literacy

I think there is a missing section give the comprehensive and ambitious title: Impact in medical informatics (not just research in general). There are huge potentials, but also significant dangers in how these models will affect medical informatics and I think this area is where LLM may impact the field of medicine the most.

Pros

1. Zero-shot NLP information extraction without requiring large supervised datasets for training, allowing for translation of notes into structured clinical information for clinical and research workflows
2. Clinical reasoning and cohort identification on massive scales that previously required significant manual annotation and labor
3. 1000s of other use cases I'm sure

However, there are significant concerns with the possibility of utilization of these models as the state-of-the-art (which they currently are)

1. Proprietary nature of the model: Very little published information exists on how the model is constructed. Therefore utilization of these models in the scientific domain comes with real risks of propagation of unknown inaccuracies or bias
2. Inability to import model and run locally. The size of these models means it will be impractical for all but a few institutions to obtain local versions for informatics research. This will lead to issues in scientific disparities and 
 - a. Reproducibility: If you do not have a static model, your results may change on the next update of the model, which means academic reproducibility can not be guaranteed.
 - b. Inability to study questions with Protected Health Information (PHI). Without the ability to import a model behind a medical systems firewall, there will be very limited ability to answer questions about or for medical documentation as it would require exportation of protected health information.
3. Pay-to-play: While medical informatics has really only had computational costs related to model training, with big companies like Meta and Google releasing their previous groundbreaking models for open use, OpenAI has set a precedent for pay-to-use model that will likely become the norm. Now every experiment that wishes to use the state-of-the-art may require payment to model developer, which may serve to limit the number of reproducibility and robustness experiments people are willing to do.

Minor comments not needing response

Computer programming

I think it's worth envisioning these tools as a way of democratizing medical informatics. Many clinicians have worthwhile hypothesis questions driven by insightful experience and knowledge about their subject matter. Often these questions, or at least the first steps, can be interrogated with relatively simple computational techniques. However, the activation barrier required to implement these first few lines of code is insurmountable. While they still may struggle a bit at more complex programming tasks, they have shown strong and consistent ability to generate code for simple questions in statistics, informatics, and figure generation.

Given the audience, it may be worth considering how this tool could be used to break in an untapped potential for scientific inquiry from clinicians and non-computational researchers to start initial hypothesis testing.

Medical education

Conscious use: I envision it can also provide direct feedback during diagnostic reasoning education, providing valuable "reps" that currently can only be gained through direct interaction with more experienced physicians. I also think it may be able to identify unexpected causal relationships in disease etiology that even master clinicians may not immediately appreciate.

I think the biggest concern for me is externalization of reasoning and I think it is important to make the distinction between this and externalization facts. Search engines started the process of easy externalization of medical knowledge, allowing for rapid and precise recall for many medical obscurities and details that previously required rote memorization. I would argue that for many reasons, that isn't necessarily detrimental, as a medical catalog of information will always be more accurate than internal memory, as long as it is easily accessible

The new possibility of LLMs is externalization of medical reasoning, with similar rapidity of retrieval. This is new territory and feels worthy of contemplation, or at least recognition.

Point-by-point response

Reviewer #1:

This research is a worthwhile effort, but the depth of analysis, language used (and novelty) would need to be improved. The paper needs a deeper analysis for a systematic overview (a Google Scholar search for "ChatGPT" "Medicine" returns 389 results for 2023 alone .. just some of the aspects that are not covered: Ethics, Translational medicine, the possibility of Copyright issues) and the language used would need to be more domain specific for a LLM, not so much "layperson"; for example, terms such as generative AI models, downstream tasks, Reinforcement Learning from Human Feedback should be used. **[R1.1]**

The authors would like to thank Reviewer #1 for their constructive feedback.

- 1. We agree that the aspects of Ethics, Translational Medicine, and potential copyright issues, had not been covered in the original version of the manuscript. Accordingly, we added a section "Ethical Use and Misinformation" in the "Patient Care" section, and further discuss the topic in our conclusion.*
- 2. Given the audience will be clinicians who may not be familiar with technical terms, we aimed to retain layperson language but added some more technical terms in brackets.*

Also, at least one very similar article is available online as preprint on medXiv, namely "The Utility of ChatGPT as an Example of Large Language Models in Healthcare Education, Research and Practice: Systematic Review on the Future Perspectives and Potential Limitations" (it should be noted that it was posted on Feb 21 2023, a few days after the authors submitted the paper). **[R1.2]**

Thank you very much for bringing this to our attention. We now cite this review. The authors agree that, with the unprecedented speed of publications on LLMs in medicine, systematic reviews for this topic are desperately needed. Therefore, while we agree that the title implies similar content, the goal of this publication is rather to set a perspective and contribute to the discussion on this highly relevant topic instead of giving a systematic overview on all existing publications on this topic.

Some additional comments:

- in the section Medical knowledge, it would make sense if the authors focus more on possible misinformation, as it is of high importance; **[R1.3]**

Thank you for this helpful remark. We agree that misinformation is an important risk of LLMs and have accordingly added a respective sub-section in the "Patient Care" section of our manuscript.

- Supplementary Table 1: it would make sense to have an additional column where the authors evaluate the output; **[R1.4]**

Thank you very much for this suggestion. We agree that explanations help understand the given examples and have added respective interpretations or explanations of the output in the "Use Case" column of Table 1 where needed. Further, with the emergence of GPT-4 we now integrate comparisons between the completions of GPT-3.5 and GPT-4 and comment on the diverging levels of accuracy between the two.

- Supplementary Table 1, point nr. 7 (code production and debugging) - the example provided is from a practical perspective useless - ChatGPT can produce concrete code, not just a vague description, which can not be used as it is, for example a simple google search would produce a much more practical solution. **[R1.5]**

Thank you very much. We have accordingly changed the example prompt for code production and debugging to provide a more useful example of the capabilities of LLMs in this area.

Reviewer #2:

This is an opinion paper about the future of medicine with the advent of large language models. The authors discuss some challenges and opportunities of LLMs.

In medicine, we have the opportunity to use domain specific LMs, pre-trained language models such as ClinicalBERT, SciBERT, BioBert to obtain better performance in clinical language applications (summarisation, information extraction, etc.).

A major Challenge of ChatGPT is factuality. This is currently addressed by the AI community since lack of factuality is dangerous in a clinical setting.

There is a lot of research in clinical informatics about these topics. **[R2.1]**

Thank you very much for bringing these points to our attention. We agree that lack of factuality is one of the major limitations and dangers of LLMs in a clinical setting, a point that was also mentioned by Reviewer #1. We have accordingly added a sub-section “Ethical Use and Misinformation” to the “Patient Care” section of the manuscript to discuss this topic in more detail. In addition, we acknowledge that a large amount of research was done in clinical NLP even before the advent of GPT-3.5. We have expanded this point in the introduction of our article and have discussed and cited previous efforts like ClinicalBERT, SciBERT, BioBert.

Supplementary material with use cases are very useful. A more in depth analysis and expansion of this position paper will be immensely useful. **[R2.2]**

The authors would like to thank Reviewer #2 for their positive feedback. We agree that examples and explanations help understand the potential and limitations of LLMs in medicine. In the revised version of the manuscript, we therefore provide a more detailed analysis on this topic including comparisons between GPT-3.5 and GPT-4 in the supplementary data.

Reviewer #3:

The manuscript provides an overview on the use of large language models (LLMs) in medicine, which is specially relevant given the recent appearance of ChatGPT. The manuscript focuses on applications of LLMs in medicine across three dimensions: patient care, research, and education.

The manuscript addresses an interesting and timely topic (or, even better, a 'hot topic'). The structure of the work is adequate and the overall idea is very interesting, however I have a few concerns for 2 out of the 3 main topics (patient care, research, education). The discussion for LLMs in research and to some extent also in education seem rather general and not very novel. In other words, if this paper had been about the application of LLMs in the financial sector, these two sections would still apply without much change. In patient care, however, many interesting aspects are discussed, including conversing with the patient when the healthcare worker is unavailable, translating and/or summarizing medical data to patients in a way that medical jargon is accessible to a layperson, and helping healthcare workers better do their administrative tasks by using speech recognition and LLMs to extract structured variables from unstructured medical texts. These discussions are specific to medicine and are very important.

I suggest that the authors revise their manuscript to include the main shortcomings and promises of LLMs for research and education *in medicine* rather than in general science (see [1,2,3,4,5], some already included as citations in your manuscript, for examples of LLMs in science in general, instead of specifically in medicine). Perhaps better highlighting what contributions are specific to medicine vs general for science could be a way to contextualize your manuscript's within the literature. **[R3.1]**

The authors would like to thank Reviewer #3 for their constructive feedback and the recognition of the value of our manuscript. We changed the manuscript to better address the medicine-specific implications of LLMs, particularly in (medical) research and (medical) education. Moreover, we changed examples 9 and 11 in Table 1 to more relevant examples. Still, while we highlight the dramatic consequences that could occur due to misinformation, especially in the medical sector, we are convinced that our manuscript is of relevance not only for medical education but for all levels of education. Concerning research, we now more specifically characterize the possibilities LLMs yield especially with the vast amounts of unstructured data in clinical settings.

A few specific comments/questions:

- "Large Language Models (LLMs) are artificial intelligence (AI) tools specifically trained to interpret and generate text." -> I would not say 'interpret' since (to me) it implies critical thinking, I would rather refer to it as 'process', since this is the term typically used in the literature to refer to models that make predictions using natural language. **[R3.2]**

Thank you very much for this suggestion. We have accordingly changed the wording.

- "LLMs improve communication between healthcare staff and patients." -> I would rephrase it as "LLMs may improve communication between healthcare staff and patients", especially if there are no citations to substantiate this claim. **[R3.3]**

Thank you very much. We have accordingly changed the wording.

[1] <https://www.nature.com/articles/d41586-023-00288-7>

[2] <https://www.nature.com/articles/d41586-023-00340-6>

[3] <https://www.nature.com/articles/d41586-023-00107-z>

[4] <https://www.nature.com/articles/d41586-023-00191-1>

[5] <https://www.nature.com/articles/d41586-022-03479-w> **[R3.4]**

*Thank you for suggesting these important recent publications on LLMs in medicine and beyond.
We have integrated these references.*

Reviewer #4:

Summary

The authors present a timely perspective of the current state of Large language models (LLM) and present a vision and use cases for how such technology (given current or near term capabilities) may impact patient care and medical research. Most importantly, the paper is written with the goal of being accessible to a general audience inclusive of researchers and clinicians within medicine and the biological sciences. Given the amount of sensationalism and misinformation surrounding LLM, I believe this goal is critically important and worthy of publication. This manuscript provides much of this needed context, as well as providing a great source for references and further reading.

With this in mind, my comments below are less directed at dissecting the article's publication value, which I support, but more suggestions and perspectives on potential improvements/additional considerations

The authors would like to thank Reviewer #4 for the positive and constructive feedback. We believe that the according changes have substantially improved clarity of the manuscript.

Major Comments

I think all the use cases are worthwhile thinking about, but I think it should be made very clear that submission of protected health information to the currently available LLMs should not be advocated. While there is an option to request no data submitted be used for future model development, there is no way of verifying this assertion. Given how easy and tempting it may be to use these LLM such as ChatGPT as note writing or diagnostic reasoning tools, the possibility of information leakage into the public domain through these LLM is real and may be worth explicitly mentioning for the sake of clinician literacy **[R4.1]**

Thank you very much for this suggestion. We agree that data privacy is one of the most important factors that will need to be considered in the context of potential future LLMs that may be used in a clinical setting. To address this point that was also mentioned by Reviewers #1 and #2, we have added a subsection on "Ethical Use and Misinformation" to the "Patient Care" section of the manuscript and have further discussed the key points in the Conclusion.

I think there is a missing section give the comprehensive and ambitious title: Impact in medical informatics (not just research in general). There are huge potentials, but also significant dangers in how these models will affect medical informatics and I think this area is where LLM may impact the field of medicine the most.

Pros

1. Zero-shot NLP information extraction without requiring large supervised datasets for training, allowing for translation of notes into structured clinical information for clinical and research workflows
2. Clinical reasoning and cohort identification on massive scales that previously required significant manual annotation and labor
3. 1000s of other use cases I'm sure

However, there are significant concerns with the possibility of utilization of these models as the state-of-the-art (which they currently are)

1. Proprietary nature of the model: Very little published information exists on how the model is constructed. Therefore utilization of these models in the scientific domain comes with real risks of propagation of unknown inaccuracies or bias

2. Inability to import model and run locally. The size of these models means it will be impractical for all but a few institutions to obtain local versions for informatics research. This will lead to issues in scientific disparities and 

a. Reproducibility: If you do not have a static model, your results may change on the next update of the model, which means academic reproducibility can not be guaranteed.

b. Inability to study questions with Protected Health Information (PHI). Without the ability to import a model behind a medical systems firewall, there will be very limited ability to answer questions about or for medical documentation as it would require exportation of protected health information.

3. Pay-to-play: While medical informatics has really only had computational costs related to model training, with big companies like Meta and Google releasing their previous groundbreaking models for open use, OpenAI has set a precedent for pay-to-use model that will likely become the norm. Now every experiment that wishes to use the state-of-the-art may require payment to model developer, which may serve to limit the number of reproducibility and robustness experiments people are willing to do. **[R4.2]**

The authors would like to thank Reviewer #3 for their constructive suggestions. We agree that LLMs have the potential to initiate paradigm changes in medical informatics. We have included your suggestions in the two new sections “Ethical Use and Misinformation” and “Reproducibility”. While we agree that sizes of current LLMs will hamper local use, we also recognize that provision of Application programming interfaces (APIs) could prevent this issue. Furthermore we recognize efforts of smaller LLMs (Vicuna: An open-source chatbot impressing GPT-4 with 90% ChatGPT quality, no date; Zhang et al., 2023). Still, we agree that there is an urgent need to encourage non-commercial open source LLMs, which we now also discuss in the section “Ethical Use and Misinformation”. Furthermore we discuss recent “Pay-to-play” options*

Minor comments not needing response

Computer programming

I think it's worth envisioning these tools as a way of democratizing medical informatics. Many clinicians have worthwhile hypothesis questions driven by insightful experience and knowledge about their subject matter. Often these questions, or at least the first steps, can be interrogated with relatively simple computational techniques. However, the activation barrier required to implement these first few lines of code is insurmountable. While they still may struggle a bit at more complex programming tasks, they have shown strong and consistent ability to generate code for simple questions in statistics, informatics, and figure generation.

Given the audience, it may be worth considering how this tool could be used to break in an untapped potential for scientific inquiry from clinicians and non-computational researchers to start initial hypothesis testing. **[R4.3]**

Thank you very much for your input on this topic. We agree that LLMs' programming capabilities may help individuals without extensive programming experience and have added this aspect to the “Research” section of the manuscript (sub-section “Computer programming”).

Medical education

Conscious use: I envision it can also provide direct feedback during diagnostic reasoning education, providing valuable “reps” that currently can only be gained through direct interaction with more experienced physicians. I also think it may be able to identify unexpected causal relationships in disease etiology that even master clinicians may not immediately appreciate.

I think the biggest concern for me is externalization of reasoning and I think it is important to make the distinction between this and externalization facts. Search engines started the process of easy externalization of medical knowledge, allowing for rapid and precise recall for many medical obscurities

and details that previously required rote memorization. I would argue that for many reasons, that isn't necessarily detrimental, as a medical catalog of information will always be more accurate than internal memory, as long as it is easily accessible

The new possibility of LLMs is externalization of medical reasoning, with similar rapidity of retrieval. This is new territory and feels worthy of contemplation, or at least recognition. **[R4.4]**

Thank you very much for these insights on externalization of reasoning. We share your perspective here and have recognized this aspect in the "Education" section of the manuscript.

References for this response letter

Vicuna: An open-source chatbot impressing GPT-4 with 90% ChatGPT quality* (no date) *Vicuna: An Open-Source Chatbot Impressing GPT-4 with 90%* ChatGPT Quality*. Available at: <https://vicuna.lmsys.org/> (Accessed: 7 April 2023).

Zhang, R. *et al.* (2023) 'LLaMA-Adapter: Efficient Fine-tuning of Language Models with Zero-init Attention', *arXiv [cs.CV]*. Available at: <http://arxiv.org/abs/2303.16199>.

Reviewers' comments:

Reviewer #1 (Remarks to the Author):

The authors have addressed all comments from my initial review. Therefore, I do not have any further comments.

Reviewer #2 (Remarks to the Author):

The authors have improved the manuscript based on the reviewer's suggestions. Nevertheless, the evaluation of LLMs with respect to fairness, ethics, harm, factuality has not been properly addressed, especially given the plethora of scientific literature covering different NLP tasks. For example, patient clinician interaction can be casted as a text simplification task. I suggest the authors look at the literature (ACL anthology and Arxiv) about related papers in areas on LLMs for clinical use cases.

Reviewer #3 (Remarks to the Author):

The manuscript provides an overview on the use of large language models (LLMs) in medicine, which is especially relevant given the recent appearance of ChatGPT. The manuscript focuses on applications of LLMs in medicine across three dimensions: patient care, research, and education.

The manuscript has improved since the last review round, and after reading all the (previous) reviewers' comments and the rebuttal, I believe it is fair to say authors have incorporated the most important comments and suggestions into the new version. The manuscript now includes a more medicine-oriented discussion, and especially when research and education are the focus. Important points were added, including ethical issues / misinformation / reproducibility, and an extensive set of example use-cases in the appendix.

One point for improvement are the references. This could be due to the nature of the field and the recency of the manuscript topic, but a large number of the references in the paper are pre-prints. It is not unheard of to find pre-prints making a claim today just to find another pre-print debunking the claim two weeks later (e.g., GPT-4 scores perfectly on a standardized medical exam). With topics where there is so much hype, we should be extra careful when using pre-prints as a major source of (scientific) knowledge.

Reviewer #4 (Remarks to the Author):

The authors have sufficiently addressed my concerns and I believe this article will be a positive impact on the field

Point-by-point response

Senior Editor Comments (in manuscript file):

Thank you very much for all your insightful and considerate comments and suggestions in the manuscript file. We agree with most of your suggestions and have included the respective changes in the revised manuscript version. In particular, we have made the following changes, which we believe have substantially improved our work:

- *Title change to avoid punctuation*
- *Rephrasing of the first paragraph of the introduction to make it more accessible to readers with various backgrounds*
- *Addition of examples (First Derm, Pahoma) to the section “Patient Care”*
- *Relocation of the section “Ethical Use and Misinformation” from “Patient Care” towards the end of the manuscript, just before the section “Conclusions and Outlook”*
- *Inclusion of a glossary of about 25 AI-related terms (Box 1)*
- *Relocation of Supplementary Table 1 to a spreadsheet to be included as Supplementary Data. Three example outputs of GPT-3.5 and GPT-4 were retained in the main manuscript (Figure 2) for better illustration of the*
- *Update of references with respect to referring to journal publications of preprint articles and inclusion of DOIs for remaining preprint articles*

Regarding your suggestion for an initial section that offers a comprehensive overview of various LLMs and their unique features, we believe the brief overview currently provided effectively grants readers an introductory understanding of the prevailing models. Given the rapid pace of evolution in this field, a detailed analysis highlighting current applications could swiftly become obsolete. As such, we would prefer to maintain our emphasis on the critical review rather than expanding into an exhaustive overview of all existing LLMs.

Reviewer #1:

The authors have addressed all comments from my initial review. Therefore, I do not have any further comments.

Thank you very much. We appreciate your input throughout the peer-review process and believe that your feedback has substantially improved our work.

Reviewer #2:

The authors have improved the manuscript based on the reviewer's suggestions. Nevertheless, the evaluation of LLMs with respect to fairness, ethics, harm, factuality has not been properly addressed, especially given the plethora of scientific literature covering different NLP tasks. For example, patient clinician interaction can be casted as a text simplification task. I suggest the authors look at the literature (ACL anthology and Arxiv) about related papers in areas on LLMs for clinical use cases.

We appreciate the reviewer's valuable feedback and agree that the evaluation of LLMs regarding fairness, ethics, harm, and factuality is a critical aspect that needs appropriate emphasis in our

manuscript. Therefore, we have moved the section “Ethical Use and Misinformation” from the “Patient Care” section towards the end of the manuscript as an individual part of the manuscript, given that its content applies to all addressed medical fields (patient care, medical research, medical education).

Moreover, following your suggestion, we have reviewed the scientific literature on LLMs, including sources such as the Association for Computational Linguistics (ACL) anthology. By incorporating this knowledge, we believe that our manuscript adequately covers the ethical considerations and evaluation frameworks related to LLMs in the context of medical applications. We appreciate your elaboration on patient-clinician interactions that can be framed as text simplification tasks and have included this tangible example in the “Patient Care” section of the manuscript.

Reviewer #3:

The manuscript provides an overview on the use of large language models (LLMs) in medicine, which is especially relevant given the recent appearance of ChatGPT. The manuscript focuses on applications of LLMs in medicine across three dimensions: patient care, research, and education.

The manuscript has improved since the last review round, and after reading all the (previous) reviewers' comments and the rebuttal, I believe it is fair to say authors have incorporated the most important comments and suggestions into the new version. The manuscript now includes a more medicine-oriented discussion, and especially when research and education are the focus. Important points were added, including ethical issues / misinformation / reproducibility, and an extensive set of example use-cases in the appendix.

One point for improvement are the references. This could be due to the nature of the field and the recency of the manuscript topic, but a large number of the references in the paper are pre-prints. It is not unheard of to find pre-prints making a claim today just to find another pre-print debunking the claim two weeks later (e.g., GPT-4 scores perfectly on a standardized medical exam). With topics where there is so much hype, we should be extra careful when using pre-prints as a major source of (scientific) knowledge.

We very much appreciate your helpful input, which we believe has substantially improved our manuscript. Following your recommendation, which was also brought up by the Senior Editor, we have double-checked the references and now cite journal publications instead of preprints wherever possible.

Reviewer #4:

The authors have sufficiently addressed my concerns and I believe this article will be a positive impact on the field

Thank you very much for your thorough peer-review and the helpful suggestions and comments. We appreciate your efforts in improving this manuscript.

REVIEWERS' COMMENTS:

Reviewer #2 (Remarks to the Author):

The authors have addressed my comments.